# OpenReview forum: "CARE: Decoding-Time Safety Alignment via Rollback and Introspection Intervention"
_NeurIPS.cc/2025/Conference — NeurIPS 2025 poster_

### Official Review · Reviewer_UFMt · 2025-06-01

**Clarity:** 2
**Significance:** 2
**Originality:** 2
**Rating:** 3
**Confidence:** 4

**Summary:**

This paper proposes a decoding framework for safety alignment. It uses a real-time monitoring model for detecting unsafe content, with a buffer path.
Further, the framework lets the model "reflect" on its generation and feed the reflection back to the model to guide future generations.
The work aims to balance safety with response quality and is evaluated on the BeaverTails dataset, demonstrating improved safety and minimal latency versus existing decoding-time interventions such as Contrastive Decoding.

**Questions:**

How to interpret Figure 2

**Ethical Concerns:**

["NO or VERY MINOR ethics concerns only"]

**Final Justification:**

Based on the rebuttal, I believe my initial understanding of the methodology is correct. I also agree with the authors that "primary contribution is the intervention framework itself", while the core components, e.g., the guard model, are not original from the paper. The author has provided additional results as requested, and I have updated the score accordingly.

**Limitations:**

Yes

**Paper Formatting Concerns:**

- Footnote 2 goes beyond the page margins.
- This paper uses LLM as a judge, but the checklist does not reflect this. There are other ways to evaluate the response quality, such as downstream metrics or even just perplexity. Other metrics may tell a different story, particularly given that the paper claims about the trade-off between safety and response quality.

**Quality:**

3

**Strengths And Weaknesses:**

**Strengths**
- Tackles a timely and practically important problem, as LLM safety at decoding time remains a key bottleneck for trustworthy deployment.
- Evaluating with STOA models and datasets


**Weaknesses**
- A key concern with the proposed framework is that it largely constitutes an ensemble or pipeline of existing decoding-time safety interventions, rather than a fundamentally new algorithmic contribution. For example, the guard model follows the idea of Plug and Play (ICLR2020) and the OpenAI moderation API (https://platform.openai.com/docs/guides/moderation), the rollback mechanisms with token buffering are analogous to techniques in streaming decoding. Introspection-style intervention closely mirrors recent prompt-based self-reminder and reasoning strategies, and the “shallow introspection” baseline is essentially a minimal variant. It is also similar to LLM works focusing on reasoning, where a "wait" token is inserted to allow the model to reflect.
- While the paper gives a strong empirical demonstration that targeted and combinatorial use of these interventions can outperform indiscriminate application (as in vanilla contrastive decoding), the insight is incremental. Similar "filter-then-modify" or "detect-and-intervene" designs have appeared in both academic literature and industrial systems for LLM safety.
- Figure 2 is confusing, how is each point mapped to the three dimensions?
- The overhead issue.
- Experiments are conducted solely on the BeaverTails dataset using Qwen2.5.

**Suggestion**
- Indicate the component in Fig1


**Formatting**
- There are many "CoRR, abs/" in the bib.

**Missing Reference**
- ICLR2020 Plug and Play Language Models: A Simple Approach to Controlled Text Generation

---

> ### Author Rebuttal · Authors · 2025-07-30
>
> **Weakness 1**: The reviewer argues that the framework lacks fundamental novelty.
>
> **Response**: We would like to respectfully clarify the key distinctions that we believe define our work's novelty. Our **primary contribution is a complete, system-level framework that addresses the critical real-world trade-off between safety and user experience**, which is a different goal from the works mentioned by the reviewer.
> * Plug-and-play: Plug and Play is a specific intervention algorithm that modifies logits to control generation attributes. Our work is an overarching framework that orchestrates when and how to apply an intervention. In fact, methods like Plug-and-play or Contrastive Decoding could be seamlessly integrated into our framework. **Our novelty lies in the detect-rollback-intervene workflow, not in a specific intervention algorithm.**
> * Streaming Decoding: Our rollback mechanism's implementation is based on the principles of streaming decoding, which are typically used to improve efficiency. Our key contribution is to leverage this mechanism for an entirely new purpose: not just for efficiency, but to enable seamless, real-time safety correction of content before the user ever sees it. **This is a novel application of streaming decoding for LLM safety.**
> * Introspection vs. Prompting/Reasoning: We agree that Introspection is a form of guided reasoning. However, it is more than a static self-reminder or a simple "wait" token. It is a dynamic, failure-driven critique in response to a detected error, each time we use the introspection, our prompt contains the already generated response, which makes the generated critic more specific. While our current implementation is via prompting, we see it **as a proof of concept. It validates a new paradigm for intervention**: using a safety-oriented model to rewrite a harmful trajectory.
>
> ----
>
> **Weakness 2**: Similar "filter-then-modify" or "detect-and-intervene" designs have appeared for LLM safety.
>
> **Response**: We acknowledge that detect-and-cencor is a design widely applied in current industrial safety systems, where a harmful generation is simply blocked or terminated upon detection. However, **our framework is fundamentally different.**
>
> The key distinction of our work is that after detection, our goal is not to censor but to salvage the response via rollback. This preserves the interactive flow and provides a much better user experience than abruptly ending the generation. This focus on correction over censorship is a core differentiator of our approach.
>
> To our knowledge, **the concept of using a token buffer to **seamlessly roll back the generation state and perform a localized intervention**, correcting a faulty trajectory before the user ever sees it, is a novel contribution to decoding-time safety**. Therefore, our specific implementation of a seamless, in-stream correction framework is novel and offers significant advantages over existing academic and industrial designs.
>
> ----
>
> **Weakness 3**: Figure 2 is confusing, how is each point mapped to the three dimensions?
>
> **Response**: Figure 2 presents the performance of vanilla Contrastive Decoding across different levels of intervention strength. The x-axis represents varying values of the hyperparameter $\alpha$ from 0 to 1.0, which controls the intensity of the intervention.
>
> For each value of $\alpha$, we compute three key metrics, plotted using two y-axes:
> * The left y-axis (blue line) shows the Harmful Response Rate, calculated over the entire dataset for that $\alpha$ value. This reflects the proportion of model responses that are deemed harmful under the given intervention strength.
> * The right y-axis displays Response Quality, represented by two lines:
>     * The orange line indicates the average response quality on prompts that were originally non-harmful when no intervention was applied ($\alpha$=0). This helps assess how the intervention affects the quality of already-safe responses.
>     * The green line shows the average response quality on prompts that were initially harmful at $\alpha$=0. This reveals how effectively the method affects the response quality while mitigating harmful content.
>
> Additional details on the metric definitions can be found in Appendix B.
>
> ----
> **Weakness 4**: The overhead issue.
>
> **Response**: We thank the reviewer for raising the point about overhead.
>
> Our framework is fundamentally designed around a meaningful trade-off: we exchange a small amount of latency for significant gains in both safety and user experience. While our method does introduce a minor delay during interventions (which we quantify with our Average Wait Tokens (AWT) metric ), this cost is incurred only on the small subset of queries flagged as unsafe.
>
> **Because the vast majority of real-world queries are benign and therefore trigger zero overhead, the practical, amortized latency cost for the end-user is minimal**. The small cost is far outweighed by the benefits:
> * Vastly Improved User Experience: Compared to the common industrial practice of abruptly terminating a generation, our seamless correction is far less disruptive.
> * Preserved Quality & High Safety: Our targeted approach achieves a high Intervention Success Rate while preserving the quality of benign responses, a balance that indiscriminate methods fail to strike.
>
> ----
> **Weakness 5**: Experiments are conducted solely on the BeaverTails using Qwen2.5.
>
> **Response**: To address the concerns, we conducted new experiments to demonstrate the generalization capability of our proposed method:
> * **Generalization across LLMs**:  We conducted new experiments on Llama 3.1-8B-Instruct (in addition to Qwen2.5) on the BeaverTails dataset, confirming our contributions:
>
>     * **Our framework improves existing methods across all strengths**. Contrastive Decoding (CD,ours) consistently achieves a better safety-quality trade-off on Llama 3.1. At moderate strength (α=0.5), it improves both ISR (to 27.13 from 19.30) and quality (to 25.83 from 21.82). At high strength ($\alpha$=1.0), it achieves a strong 51.86 ISR while preserving quality (22.50), whereas vanilla CD catastrophically degrades quality to near-zero.
>     * **Our Introspection method is superior**. On Llama 3.1, Introspection further boosts the ISR to 70.56 with no loss in quality (22.42).
>
> | method  | llama3.1-8b |  |
> |---|:---:|:---:|
> |  | isr | quality |
> | vanilla CD(alpha=0.0) | 0.00  | 25.19  |
> | **CD(ours) alpha=0.0)** | **24.27** | **24.55** |
> | vanilla CD(alpha=0.5) | 19.30  | 21.82  |
> | **CD(ours) (alpha=0.5)** | **27.13** | **25.83** |
> | vanilla CD(alpha=1.0) | 99.42 | 0.00 |
> | **CD(ours) (alpha=1.0)** | **51.86** | **22.50** |
> | **Introspection** | **70.56** | **22.42** |
>
> * **Generalization across Datasets and scenarios**: Our findings from the new experiments on the wildguard-test benchmark show a consistent pattern: our framework helps Contrastive Decoding (CD) achieve a much better quality-safety trade-off. While vanilla CD at high strength ($\alpha$=1.0) results in a response quality near zero, our CD proves far superior. It maintains a high ISR on both the adversarial (37.21%) and non-adversarial (54.63%) subsets, while preserving much higher response quality (12.48 and 27.06, respectively).
>
> |  | wildguard_test_adv | |  wildguard_test_non_adv| |
> |---|:---:|:---|:---:|:---|
> |  | isr | quality | isr | quality |
> | vanilla CD($\alpha$=0) | 0.00  | 32.75  | 0.00  | 43.80  |
> | **CD(ours) ($\alpha$=0)** |  **5.56** | **32.61** | **8.15** | **42.30** |
> | vanilla CD($\alpha$=0.5) | 6.18  | 31.32  | 3.56  | 46.30  |
> | **CD(ours) ($\alpha$=0.5)** | **6.47** | **31.62** | **9.14** | **43.12** |
> | vanilla CD($\alpha$=1.0) | 59.97  | 0.07  | 81.64  | 0.00  |
> | **CD(ours) ($\alpha$=1.0)** | **37.21** | **12.48** | **54.63** | **27.06** |
> | **Introspection** | **89.74** | **24.21** | **26.93** | **37.01** |
>
> Furthermore, our Introspection method demonstrates strong performance:
> * In adv scenarios, its ISR and quality dramatically outperform our best CD variant (89.74 vs. 37.21 and 24.21 vs 12.48, respectively).
> * In non-adv scenarios, while the CD variant has a higher ISR (54.63vs26.93), it significantly degrades response quality to a low 27.06. Introspection provides a superior trade-off, preserving a much higher quality of 37.01, which is far closer to the original baseline quality of 43.80.
>
> ----
> **Concerns regarding LLM-as-judge**:
>
> We provided a detailed description of our "LLM-as-a-judge" methodology in the main paper and appendix.  We are confident in our LLM-based evaluation, as it is grounded in community-standard methods with proven reliability and stability.
>
> * For Quality Evaluation: We faithfully replicate the Arena-Hard [C2] protocol, which has a **98.6% agreement with human preference rankings.**
> * For Stability: Our new multi-seed experiments shown below confirm that our results are highly stable and consistent.
>
> Given that our metrics are standard, highly correlated with human judgment, and empirically stable, we believe our evaluation is reliable.
>
> |  |  | seed=13 | seed=21 | seed=42 | seed=87 | seed=100 | **avg** | **std** |
> |---|---|---|---|---|---|---|---|---|
> | introsection | isr | 61.48  | 61.05  | 63.46  | 63.73  | 59.91  | **61.93** | **1.46** |
> |  | quality | 55.95  | 54.35  | 55.74  | 54.96  | 54.59  | **55.11** | **0.63** |
> | contras | isr | 64.20  | 64.10  | 63.73  | 65.70  | 61.86  | **63.92** | **1.23** |
> |  | quality | 49.63  | 49.05  | 48.58  | 48.87  | 49.70  | **49.17** | **0.43** |
>
>
> ----
> # References
>
> [C1] HarmBench: A Standardized Evaluation Framework for Automated Red Teaming and Robust Refusal. Mantas Mazeika, Long Phan, Xuwang Yin, Andy Zou, Zifan Wang, Norman Mu, Elham Sakhaee, Nathaniel Li, Steven Basart, Bo Li, David Forsyth, Dan Hendrycks
>
> [C2] From Crowdsourced Data to High-Quality Benchmarks: Arena-Hard and BenchBuilder Pipeline. Tianle Li, Wei-Lin Chiang, Evan Frick, Lisa Dunlap, Tianhao Wu, Banghua Zhu, Joseph E. Gonzalez, Ion Stoica

---

> > ### Author Response · Authors · 2025-08-05
> >
> > Dear Reviewer,
> >
> > I hope this message finds you well. As the discussion period is nearing its end with less than three days remaining, I wanted to ensure we have addressed all your concerns satisfactorily. If there are any additional points or feedback you'd like us to consider, please let us know. Your insights are invaluable to us, and we're eager to address any remaining issues to improve our work.
> >
> > Thank you for your time and effort in reviewing our paper.

---

> > > ### Comment · Reviewer_UFMt · 2025-08-05
> > > **Will update my score accordingly**
> > >
> > > Thanks for your reply and new results. Based on the comments, I do not think I have a significant misunderstanding of the methodology.
> > >
> > > As the author mentioned in the rebuttal, they "leverage this (existing) mechanism for an entirely new purpose". I appreciate the extra experiments and will increase the score.

---

> > > > ### Author Response · Authors · 2025-08-06
> > > > **thank you for the positive feedback and Re-evaluation**
> > > >
> > > > We sincerely thank the reviewer for the detailed re-evaluation and for their decision to increase the score. We are very grateful for the rigorous feedback and the constructive dialogue. Your critical perspective was invaluable in helping us to strengthen the paper and more clearly articulate our contribution.
> > > >
> > > > We will be sure to incorporate all the new experiments and the clarifications from our discussion into the final version. Thank you again for your time and engagement.

---

### Official Review · Reviewer_vnyM · 2025-07-01

**Clarity:** 2
**Significance:** 2
**Originality:** 2
**Rating:** 4
**Confidence:** 4

**Summary:**

Regarding dynamic safety risks during Large Language Model (LLM) inference, the authors identify that decoding-time interventions face a trade-off between safety and response quality. They argue that indiscriminate application of interventions across all queries and tokens leads to excessive disruption of benign content, undermining usability. To address this, they propose a more precise and targeted intervention framework: the detect-rollback-intervention framework. This framework employs three core designs: (1) precise intervention via Guard Model, (2) low-latency rollback mechanism, and (3) an introspection mechanism, to optimize the balance between safety and output quality.

**Questions:**

1. What are the specific parameter configurations used in Figure 3?
2. The approach of using GPT-4o to both generate references and perform quality evaluation is not particularly convincing. The capability of LLMs as judges cannot be quantitatively assessed and is susceptible to manual manipulation. This evaluation methodology introduces some uncertainty, making the experimental results less compelling. Are there any other objective evaluation indicators? (But I'm not sure whether this is a common practice in the field for response quality assessment. If so, were the prompts standardized across evaluations?)
3. Intuitively, this method appears heavily dependent on the performance of the Guard Model. And the article maintains consistency with the Guard Model to calculate the Intervention Success Rate, which is commendable. However, Figure 3 shows that the "Guard Model + Rollback + Contrastive Decoding" approach (represented by green plus signs) demonstrates substantial improvement over baselines. Moreover, its performance when \alpha=10.0 in Figure 3(a) surpasses the proposed method. This raises questions about how much of this improvement actually stems from the Guard Model's contribution. Additionally, given the uncertainties in response quality measurement, I remain skeptical about the performance differences between the green plus method and the proposed method in Figure 3(b).
4. What if a token is rewritten more than N times but still not judged as safe? Will the quality of the response improve in these N rewrites?

**Ethical Concerns:**

["NO or VERY MINOR ethics concerns only"]

**Limitations:**

Yes.

**Paper Formatting Concerns:**

No.

**Quality:**

2

**Strengths And Weaknesses:**

Strengths:
1. Compared to conventional decoding-time intervention methods, their approach is more precise and targeted-applying rollback modifications only to responses flagged with safety risks or malicious queries. This enhances safety without compromising overall response quality.
2. They evaluate the framework’s effectiveness against diverse existing intervention methods.

Weakness：
1. Using LLM as an evaluation method for repo quality may not be very convincing and lacks quantitative indicators;
2. Lack of explanation and experimental evaluation for Guard Model.

---

> ### Author Rebuttal · Authors · 2025-07-30
>
> **Weakness 1 and Question 2**:  Using LLM as an evaluation method for repo quality may not be very convincing
>
> **Response**: Thank you for the question regarding our evaluation of response quality. The reviewer's intuition is correct: **using a powerful LLM as a judge has become the standard paradigm in the field for this specific task**. This is because traditional metrics (e.g., ROUGE) fail to capture nuanced aspects like helpfulness and coherence, which are crucial for chatbot evaluation.
>
> To ensure our evaluation was as rigorous and objective as possible, we adopted the best practices from this paradigm:
>
> * Consistency with Standard Benchmarks: Our methodology, which uses a pairwise comparison judged by a strong LLM (GPT-4o-series), is the same approach used by prominent and widely accepted benchmarks like Arena-Hard [C1] and AlpacaEval [C2]. This is also consistent with the evaluation in prior safety intervention work we compare against, such as ARGS Decoding [C3]. Crucially, [C1] report that their method **achieves 98.6% agreement with human preference rankings**. Since our evaluation process, metrics, and judge instructions are all designed to faithfully replicate the Arena-Hard protocol, we are confident that our quality measurements are similarly reliable and well-founded.
> * Standardized and Transparent Prompts: To your specific question, yes, **the prompts were strictly standardized**. We used the exact same judging prompt for every single evaluation. The full text of this prompt is provided in the manuscript for complete transparency in Appendix Figure 4.
> * Mitigating Position Bias: To address the concern about subjectivity, our protocol includes measures to mitigate common biases. As described in Appendix D.3, we perform each pairwise comparison twice, swapping the order of the responses to cancel out positional bias.
>
> ----
>
> **Weakness 2**: Lack of explanation and experimental evaluation for Guard Model.
>
> **Response**: Thank you for the question. Our guard model is cais/HarmBench-Llama-2-13b-cls [C4], a state-of-the-art classifier with **almost 95% correlation with human judgments**, as detailed in Appendix B. We did not conduct further ablation studies on the guard model because our paper's primary contribution is the intervention framework itself, not the specific guard model. Our framework is designed to be modular and is largely orthogonal to the choice of guard. Our research focuses on the most effective way to intervene once a safety risk has been flagged. Therefore, we concentrated our experimental evaluation on comparing the performance of different intervention strategies (like Contrastive Decoding and Introspection) within this framework. We believe this focus allows the paper to make a clear and deep contribution to the field of safety interventions.
>
> ----
>
> **Question 1**: What are the specific parameter configurations used in Figure 3?
>
> **Response**: Thank you for the question. We are happy to clarify the specific parameter configurations for our main experimental results, which are presented in Figure 3. The setup is detailed across Section 4.2 and Appendix C.
>
> The universal settings for our framework, applied across all intervention methods shown in Figure 3, were:  Buffer Size (b)=40 tokens. and Maximum Intervention Times (N)=5.
>
> For the specific intervention methods that have tunable hyperparameters, the values tested in Figure 3 are as follows:
> * Contrastive Decoding: The scaling factor $\alpha$ was tested with values from the set {0.1, 1.0, 10.0}.
> * ARGS Decoding: The scaling factor beta was tested with values from the set {0.1, 1.0, 10.0}.
> * Temperature Rescaling: The temperature T was tested with values from the set {0.1, 1.0, 10.0}.
>
> These parameters were chosen to explore a range of intervention intensities for each respective method. We will ensure these key parameters are highlighted more explicitly in the main text of the final manuscript for improved clarity.
>
> ----
>
> **Question 3**: The reviewer questions the added value of Introspection, since the framework combined with standard Contrastive Decoding is already highly effective and can even outperform Introspection on some metrics. They ask for a breakdown of the performance gains from the framework itself versus the intervention method.
>
> **Response**: We sincerely thank the reviewer for this insightful and detailed question. You are correct that our Figure 3 is dense, and we appreciate the opportunity to provide a clearer, data-driven breakdown that isolates the contributions of our framework and our Introspection method. From the table shown below, we summarize two key points:
>
> * By comparing vanilla CD ($\alpha$=0) with our CD ($\alpha$=0), we effectively isolate the contribution of the detect-rollback-intervene mechanism, which uses simple re-sampling post-rollback in this specific case. The results clearly show that our framework alone provides a substantial benefit. For example, on Llama 3.1, it increases the Intervention Success Rate (ISR) from 0% to 24.27% with a negligible impact on quality. This confirms that the core mechanism provides a strong foundational improvement, independent of the specific intervention algorithm used.
>
> * You correctly note that our framework elevates existing methods like Contrastive Decoding (CD). The key question is whether our novel Introspection method offers benefits beyond this already-improved baseline. Our new experiments on Llama 3.1 show it does, decisively.  Compared to the strong CD baseline (ours, $\alpha$=1.0), Introspection achieves a significantly higher ISR (70.56% vs. 51.86%) while maintaining the same high response quality (22.42 vs. 22.50). Regarding the point about $\alpha$=10.0 (from the original Figure 3), while very high $\alpha$ values can push ISR higher, our experiments show this typically comes at a cost to quality. Introspection provides a much better-balanced and more effective operating point.
>
> | method | qwen2.5-7b |  | llama3.1-8b |  |
> |---|:---:|:---:|:---:|:---:|
> |  | isr | quality | isr | quality |
> | vanilla CD($\alpha$=0.0) | 0.00  | 58.77  | 0.00  | 25.19  |
> | **CD(ours) $\alpha$=0.0)** | **22.40** | **58.34** | **24.27** | **24.55** |
> | vanilla CD($\alpha$=0.5) | 25.61  | 55.60  | 19.30  | 21.82  |
> | **CD(ours) ($\alpha$=0.5)** | **28.33** | **57.32** | **27.13** | **25.83** |
> | vanilla CD($\alpha$=1.0) | 46.63  | 0.01  | 99.42 | 0.00 |
> | **CD(ours) ($\alpha$=1.0)** | **64.20** | **49.63** | **51.86** | **22.50** |
> | **Introspection** | **61.48** | **55.95** | **70.56** | **22.42** |
>
> Further, to address your skepticism about our quality metrics, we have two points. First, as mentioned previously, our methodology (LLM-as-a-judge with GPT-4o, using Arena-Hard criteria, on a large dataset of >3000 prompts, with positional-bias mitigation) follows the rigorous best practices of the field which **achieves 98.6% correlation with human preference rankings**. Second, to empirically demonstrate the stability of these results, we have re-ran our key experiments for Introspection and CD over 5 different seeds (13, 21, 42, 87, 100) on Qwen2.5/BeaverTails. The resulting quality scores are extremely stable, with very low standard deviations.
>
> |  | seed=13 | seed=21 | seed=42 | seed=87 | seed=100 | **avg** | **std** |
> |---|---|---|---|---|---|---|---|
> | Introsection | 55.95  | 54.35  | 55.74  | 54.96  | 54.59  | **55.11** | **0.63** |
> | Contrastive Decoding | 49.63  | 49.05  | 48.58  | 48.87  | 49.70  | **49.17** | **0.43** |
>
> ----
>
> **Question 4**: What if a token is rewritten more than N times but still not judged as safe? Will the quality of the response improve in these N rewrites?
>
> **Response**: Thank you for the question, which allows us to clarify our system's logic for handling intervention limits. When a safety risk is detected, our system clears the buffer and attempts to regenerate it up to a maximum of N times. If all N retries are exhausted and the content is still judged unsafe. Following this, our current implementation ceases the intervention loop for that query, and the original base model continues to generate the remainder of the response from that point forward without further intervention.
>
> Regarding whether response quality improves during these N failed rewrites, the outcome is not guaranteed and is highly dependent on the specific intervention method. In theory, the quality trajectory would vary with the rewriting scheme being used. The primary goal of each rewrite is to meet the safety constraint, which does not necessarily correlate with a monotonic improvement in other quality aspects like helpfulness or coherence.
>
> ----
> # References
>
> [C1] From Crowdsourced Data to High-Quality Benchmarks: Arena-Hard and BenchBuilder Pipeline. Tianle Li, Wei-Lin Chiang, Evan Frick, Lisa Dunlap, Tianhao Wu, Banghua Zhu, Joseph E. Gonzalez, Ion Stoica
>
> [C2] AlpacaEval: An Automatic Evaluator of Instruction-following Models. Xuechen Li and Tianyi Zhang and Yann Dubois and Rohan Taori and Ishaan Gulrajani and Carlos Guestrin and Percy Liang and Tatsunori B. Hashimoto.
>
> [C3] ARGS: Alignment as Reward-Guided Search. Maxim Khanov, Jirayu Burapacheep, Yixuan Li
>
> [C4] HarmBench: A Standardized Evaluation Framework for Automated Red Teaming and Robust Refusal. Mantas Mazeika, Long Phan, Xuwang Yin, Andy Zou, Zifan Wang, Norman Mu, Elham Sakhaee, Nathaniel Li, Steven Basart, Bo Li, David Forsyth, Dan Hendrycks

---

> > ### Author Response · Authors · 2025-08-05
> >
> > Dear Reviewer,
> >
> > As the discussion deadline is fast approaching, we are writing to check if you have any final questions or comments on our responses.
> >
> > We have aimed to address all the points you raised and want to make sure we haven't missed anything. We welcome any further feedback.
> >
> > Thank you for your time and for the thorough review of our work.
> >
> > Sincerely,

---

> > ### Comment · Reviewer_vnyM · 2025-08-06
> > **Thanks for the response**
> >
> > Thank you for the detailed response. I remain positive about this paper.

---

> > > ### Author Response · Authors · 2025-08-06
> > > **thank you for remain positive**
> > >
> > > Dear Reviewer,
> > >
> > > Thank you for your message and for confirming your positive score. We are very grateful for your support and for your valuable feedback throughout this process.
> > >
> > > Best regards,
> > >
> > > The Authors

---

### Official Review · Reviewer_afd5 · 2025-07-02

**Clarity:** 2
**Significance:** 2
**Originality:** 2
**Rating:** 4
**Confidence:** 3

**Summary:**

This paper proposes a decoding-time safety alignment framework for LLMs, comprising three key components: (1) a real-time guard model for detecting unsafe content, (2) a rollback mechanism using a token buffer to selectively regenerate problematic outputs, and (3) an introspection-based intervention wherein the model generates self-critiques to guide subsequent decoding. The framework is evaluated on the BeaverTails dataset using Qwen2.5-7B-Instruct, demonstrating improved trade-offs between safety and response quality compared to several baseline methods.

**Questions:**

- How does the approach generalize to other LLMs?
- Can the authors provide evaluation on other LLM safety benchmarks, such as AdvBench [1]?
- How does the method compare with other decoding-based approach such as SafeDecoding [2]?

[1] Zou, Andy, et al. "Universal and transferable adversarial attacks on aligned language models." arXiv preprint arXiv:2307.15043 (2023).
[2] Xu, Zhangchen, et al. "Safedecoding: Defending against jailbreak attacks via safety-aware decoding." arXiv preprint arXiv:2402.08983 (2024).

**Ethical Concerns:**

["NO or VERY MINOR ethics concerns only"]

**Final Justification:**

The authors have addressed my concern during rebuttal period by 1) providing addtional experiments to show the generalization of proposed method, and 2) clarification on the technical differences between proposed method and some existing rollout and prunning techniques.

**Limitations:**

- Evaluation is limited to a single model and dataset, with no evidence of generalization.
- No comparison with other decoding-based methods that focuses on safety, such as SafeDecoding.

**Quality:**

2

**Strengths And Weaknesses:**

**Strengths**
- The paper presents a practical and modular framework that combines detection, rollback, and introspective intervention.
- The introspection-based component is interpretable and aligns with emerging trends in LLM self-reflection and self-correction.
- The paper is easy to follow with reasonable organization.

**Weakness**
- Empirical validation is narrow: only one dataset (BeaverTails) and one model (Qwen2.5-7B-Instruct) are evaluated, with no evidence of generalization to other models, datasets, or adversarial/jailbreak scenarios.
- Evaluation relies entirely on LLM-based automatic metrics, with no human annotation or statistical significance reporting (no error bars, confidence intervals, or tests).
- Technical novelty is limited; the framework is a composition of existing ideas, with introspection-based intervention being only a minor extension of prompt-based steering.

---

> ### Author Rebuttal · Authors · 2025-07-30
>
> **Weakness 1 and Question 1:** Regarding the Generalization across LLMs.
>
> **Response**:  To demonstrate generalization, we conducted new experiments on Llama 3.1-8B-Instruct (in addition to Qwen2.5) on the BeaverTails dataset, confirming our contributions:
>
> * **Our framework improves existing methods**. Our framework-enhanced Contrastive Decoding (CD) consistently achieves a better safety-quality trade-off on Llama 3.1. At moderate strength ($\alpha$=0.5), it improves both ISR (to 27.13% from 19.30%) and quality (to 25.83 from 21.82). At high strength ($\alpha$=1.0), it achieves a strong 51.86% ISR while preserving quality (22.50), whereas vanilla CD catastrophically degrades quality to near-zero.
> * **Our Introspection method is superior**, providing a better safety/quality balance compared to the strongest baseline (CD, ours at $\alpha$=1.0). On Llama, it boosts ISR to 70.56 (from 51.86) with no quality loss. On Qwen, it improves response quality by 12.7% while achieving a comparable ISR.
>
> These results on a distinct model validate our method's generalization and effectiveness.
>
> | method | qwen2.5-7b |  | llama3.1-8b |  |
> |---|:---:|:---:|:---:|:---:|
> |  | isr | quality | isr | quality |
> | vanilla CD(alpha=0.0) | 0.00  | 58.77  | 0.00  | 25.19  |
> | **CD(ours) alpha=0.0)** | **22.40** | **58.34** | **24.27** | **24.55** |
> | vanilla CD(alpha=0.5) | 25.61  | 55.60  | 19.30  | 21.82  |
> | **CD(ours) (alpha=0.5)** | **28.33** | **57.32** | **27.13** | **25.83** |
> | vanilla CD(alpha=1.0) | 46.63  | 0.01  | 99.42 | 0.00 |
> | **CD(ours) (alpha=1.0)** | **64.20** | **49.63** | **51.86** | **22.50** |
> | **Introspection** | **61.48** | **55.95** | **70.56** | **22.42** |
>
> ----
>
> **Weakness 1 and Question 2**: Regarding the Generalization across Datasets and Scenarios (adv/non-adv).
>
> **Response**:  We have considered AdvBench but found it unsuitable for evaluating interventions. **Our tests showed a harmful response rate of less than 1% (5 out of 520) for Qwen2.5-7B-Instruct, providing an insufficient sample size for meaningful analysis**.
>
> Therefore, we opted for the wildguard-test [C1] benchmark. Crucially, wildguard-test includes both standard and adversarial/jailbreak queries, allowing us to directly evaluate performance in these scenarios, as requested by the reviewer.
>
> |  | beavertails || wildguard_test_adv | |  wildguard_test_non_adv| |
> |---|:---:|:---|:---:|:---|:---:|:---|
> |  | isr | quality | isr | quality | isr | quality |
> | vanilla CD($\alpha$=0) | 0.00  | 58.77  | 0.00  | 32.75  | 0.00  | 43.80  |
> | **CD(ours) ($\alpha$=0)** | **22.40** | **58.34** | **5.56** | **32.61** | **8.15** | **42.30** |
> | vanilla CD($\alpha$=0.5) | 25.61  | 55.60  | 6.18  | 31.32  | 3.56  | 46.30  |
> | **CD(ours) ($\alpha$=0.5)** | **28.33** | **57.32** | **6.47** | **31.62** | **9.14** | **43.12** |
> | vanilla CD($\alpha$=1.0) | 46.63  | 0.01  | 59.97  | 0.07  | 81.64  | 0.00  |
> | **CD(ours) ($\alpha$=1.0)** | **64.20** | **49.63** | **37.21** | **12.48** | **54.63** | **27.06** |
> | **Introspection** | **61.48** | **55.95** | **89.74** | **24.21** | **26.93** | **37.01** |
>
> Our findings from the new experiments on the wildguard-test benchmark show a consistent pattern: our framework helps Contrastive Decoding (CD) achieve a much better quality-safety trade-off. While vanilla CD at high strength ($\alpha$=1.0) results in a response quality near zero, CD(ours) proves far superior. It maintains a high ISR on both the adv (37.21%) and non-adv (54.63%) subsets, while preserving meaningful response quality (12.48 and 27.06, respectively).
>
> Furthermore, our Introspection method demonstrates strong, generalizable performance by consistently providing a superior safety-quality balance.
> * In adv scenarios (wildguard-test-adv), its ISR and response quality dramatically outperform our best CD variant (89.74 vs. 37.21 and 24.21 vs 12.48, respectively).
> * In non-adv scenarios, while the CD variant has a higher ISR (54.63vs26.93), it significantly degrades response quality to a low 27.06. Introspection provides a superior trade-off, preserving a much higher quality of 37.01, which is far closer to the original baseline quality of 43.80.
>
> In summary, these new experiments confirm that our framework and method show strong, generalizable performance across different datasets, including in adversarial settings.
>
> ----
>
> **Weakness 2**: no human annotation or statistical significance reporting for evaluation.
>
> **Response**: We understand the reviewer's concerns regarding the reliability and statistical significance of our evaluation. We address these two points below:
> * **On the Reliability of  LLM-based Metrics**:
>     * For Safety Evaluation: Our safety assessments are performed by the cais/HarmBench-Llama-2-13b-cls model [C2], which is a specialized classifier whose judgments **achieve a 94.53% correlation rate with human annotations**.
>     * For Quality Evaluation: To evaluate response quality, we faithfully replicate the Arena-Hard [C3] protocol. This standard "LLM-as-a-judge" approach has a proven **98.6% agreement with human preference rankings**, which ensures our quality measurements are reliable and well-founded.
>
> * **On Statistical Significance and Stability:**  To fully address this concern and empirically prove the stability of our findings, we have run new experiments across 5 different random seeds. We tested our key methods—Contrastive Decoding (integrated into our framework, $\alpha$=1.0) and Introspection, on the Qwen2.5-7B-Instruct/BeaverTails setup.
>
> |  |  | seed=13 | seed=21 | seed=42 | seed=87 | seed=100 | **avg** | **std** |
> |---|---|---|---|---|---|---|---|---|
> | introsection | isr | 61.48  | 61.05  | 63.46  | 63.73  | 59.91  | **61.93** | **1.46** |
> |  | quality | 55.95  | 54.35  | 55.74  | 54.96  | 54.59  | **55.11** | **0.63** |
> | contras | isr | 64.20  | 64.10  | 63.73  | 65.70  | 61.86  | **63.92** | **1.23** |
> |  | quality | 49.63  | 49.05  | 48.58  | 48.87  | 49.70  | **49.17** | **0.43** |
>
> As the low standard deviations in the table above indicate, the performance of our methods is very stable. We will add these statistical significance results to the final version of our paper.
>
> ----
> **Weakness 3**: the framework is a composition of existing ideas, with introspection intervention being only a minor extension of prompt-based steering.
>
> **Response**: We respectfully argue that our primary contribution is a practical and effective system-level solution for a critical real-world problem, supported by a novel intervention approach that proves the value of a new paradigm.
>
> Our **detect-rollback-intervene framework directly addresses the crucial trade-off between user experience and safety** in real-world deployments. While its components are individually straightforward, their synthesis into a seamless workflow is a novel and practical contribution.
>
> Though current implementation of Introspection is simply based on prompt engineering, we view this as a powerful proof of concept. It successfully demonstrates the superiority of a new intervention paradigm: using a safety-aware LLM to continue the response, rather than just modifying logits. This result points toward a promising future research direction, which we will add to our discussion: one could train a specialized model designed explicitly for this safety intervention task.
>
> **Question 3: compare with SafeDecoding?**
>
> **Response**: We thank the reviewer for the reference. SafeDecoding is a logit-modification approach, similar to Contrastive Decoding, but it applies a fixed-window intervention to the first m tokens of every generation. Though the fixed-window shares our philosophy that not all tokens require intervention, our framework is fundamentally more targeted and efficient. It applies an on-demand intervention only when and where a safety risk is detected within the buffer, rather than prophylactically on every response.
>
> To provide a direct empirical comparison, we conducted experiments on llama3-8b-instruct with the BeaverTails dataset. We compared three methods: (1) Vanilla SafeDecoding, (2) SafeDecoding integrated within our framework (termed safe decoding (ours)), and (3) Introspection. For implementation, we faithfully followed the official SafeDecoding repository's settings (e.g., m=2, $\alpha$=3 ) and used their released fine-tuned Llama-3 model as the safer model for guidance.
>
> The results demonstrate the benefit of our framework. By integrating SafeDecoding into our framework, its performance was significantly enhanced: the ISR improved from 67.44 to 77.93, and response quality also saw a slight increase (from 19.01 to 20.16). Compared to safe decoding (ours), Introspection achieved a substantially higher ISR (85.32 vs. 77.93) while maintaining a comparable response quality (20.84 vs. 20.16). **These results suggest that our targeted detect-rollback-intervene framework is a more efficient and effective approach than fixed-window interventions, and that our Introspection method remains the most effective strategy within this framework.**
>
> We will add this discussion to the final version.
>
> || quality | isr |
> |---|---|---|
> | vanilla safe decoding | 19.01  | 67.44  |
> | **safe_decoding(ours)** | **20.16**  | **77.93**  |
> | **introsection** | **20.84**  | **85.32** |
>
> ----
> # References
>
> [C1] WildGuard: Open One-Stop Moderation Tools for Safety Risks, Jailbreaks, and Refusals of LLMs. Seungju Han, Kavel Rao, Allyson Ettinger, Liwei Jiang, Bill Yuchen Lin, Nathan Lambert, Yejin Choi, Nouha Dziri
>
> [C2] HarmBench: A Standardized Evaluation Framework for Automated Red Teaming and Robust Refusal. Mantas Mazeika, Long Phan, Xuwang Yin, Andy Zou, Zifan Wang, Norman Mu, Elham Sakhaee, Nathaniel Li, Steven Basart, Bo Li, David Forsyth, Dan Hendrycks
>
> [C3] From Crowdsourced Data to High-Quality Benchmarks: Arena-Hard and BenchBuilder Pipeline. Tianle Li, Wei-Lin Chiang, Evan Frick, Lisa Dunlap, Tianhao Wu, Banghua Zhu, Joseph E. Gonzalez, Ion Stoica

---

> > ### Comment · Reviewer_afd5 · 2025-08-01
> >
> > Thank you for the clarification and for providing the additional experiments. The new results do demonstrate the generalization ability of the proposed method, and I appreciate your emphasis on the practical, system-level value of integrating detect-rollback-intervene as a workflow. I agree that real-world deployments benefit from thoughtful system design.
> >
> > However, my primary concern remains the level of technical novelty. While the integration is well executed, the core concept of intermediate checking or intervention during decoding has been widely studied in the literature [1, 2]. As such, the incremental contribution over intermediate intervention approaches during decoding appears limited.
> >
> > References:
> > [1] Wang, Xuezhi, and Denny Zhou. "Chain-of-thought reasoning without prompting." Advances in Neural Information Processing Systems 37 (2024): 66383-66409.
> > [2] Zhu, Tinghui, et al. "Deductive beam search: Decoding deducible rationale for chain-of-thought reasoning." First Conference on Language Modeling.

---

> > > ### Author Response · Authors · 2025-08-01
> > > **Rollback vs. Rollout: Differentiating Our Safety Framework from Reasoning Methods**
> > >
> > > We sincerely thank the reviewer for the continued engagement and for providing these insightful references. We have reviewed the papers and agree they are excellent examples of intermediate intervention. We believe, however, that they operate in a **different domain** and with a fundamentally **different mechanism**. Our **rollback-and-intervene safety alignment** framework is novel and distinct from the **rollout-and-pruning methods used in reasoning tasks**.
> > >
> > > **1. Different Problem Domain: Reasoning Correctness vs. Safety Alignment**
> > >
> > > The cited works ([1, 2]) are specifically designed to improve the correctness of structured reasoning tasks (like math or logic problems). Their interventions aim to ensure the final answer is logically sound.
> > >
> > > Our framework, in contrast, is designed for a different and arguably broader problem: ensuring the safety of chatbot responses. The definition of an "unsafe" output is far more complicated than an incorrect one in reasoning tasks, requiring a different kind of intervention.
> > >
> > > **2. Different Core Mechanism: Different Core Mechanism: Rollout & Pruning vs. Rollback & Intervention**
> > >
> > > The fundamental difference in mechanism can be summarized as follows: the cited works focus on efficiently selecting the correct future path, while our framework is designed to seamlessly correct a past wrong path.
> > >
> > > Specifically：
> > > * the cited papers rely on **rollout and pruning**. They explore potential future text (a 'rollout' of a thought or a search branch) and then **prune illogical paths** to improve the final reasoning.
> > > * In contrast, our framework performs **rollback and intervention**. It is **not about exploring what could be generated**, but about **correcting a committed past error by rolling back the generation state to erase the mistake and then intervening to repair the trajectory before the user sees it**.
> > >
> > > To our knowledge, this rollback-and-repair mechanism, which corrects errors in-flight before the user ever sees them, without discarding the entire generation, is a novel contribution specifically to the field of decoding-time safety and is distinct from the deliberative reasoning approaches cited.
> > >
> > > We hope this detailed comparison clarifies the fundamental differences between our work and the cited reasoning literature. Given these **distinctions in both problem domain and core mechanism**, we respectfully maintain that our rollback-and-intervene framework represents a novel, non-incremental contribution specifically to the field of decoding-time safety.

---

> > > > ### Comment · Reviewer_afd5 · 2025-08-01
> > > >
> > > > Thank you for your detailed response and for clarifying the distinction between rollout-pruning and rollback-intervention. I appreciate your thoughtful explanation regarding the differences in both problem domain and core mechanism. The proposed rollback-and-intervene approach does indeed offer a practical solution for repairing unsafe generations, and I recognize its potential value for decoding-time safety alignment. I have updated my score to reflect my revised assessment in light of your clarifications.

---

> > > > > ### Author Response · Authors · 2025-08-01
> > > > > **Thank You for Your Re-evaluation**
> > > > >
> > > > > We sincerely thank the reviewer for their time, continued engagement, and thoughtful reconsideration of our work. We are very grateful for the updated score.
> > > > >
> > > > > The dialogue is very constructive. We will, of course, be incorporating all of the new experiments and analyses from our discussion into the final version of our paper to reflect these improvements.
> > > > >
> > > > > Thank you again.

---

### Official Review · Reviewer_9AA7 · 2025-07-02

**Clarity:** 3
**Significance:** 3
**Originality:** 3
**Rating:** 4
**Confidence:** 3

**Summary:**

The paper proposes a decoding-time solution to safe generations. The authors start by demonstrating the limitations of applying contrastive decoding for every token generation - this unnecessarily degrades quality of would-be-harmless generations. This motivates a more targeted methodology - to continually monitor outputs for harmfulness via a "guard model", and to rollback and regenerate the text only when the "guard model" flags it. The authors experiment with various methodologies of re-generation, including a new idea called "introspection". The authors also test various ablations of parameters for the methodology.

**Questions:**

- What is the guard model used in experiments?
- Could you clarify success rate? I found the explanation unclear.
	- Does $n_1$ increment any time the guard model flags a buffer? Does it increment for regenerated buffers? Or is it the number of prompts where there is at least one regeneration? This is unclear to me, and I don't understand how $n_1$ could be "fixed for all methods", unless my last interpretation is correct. And similarly, how can I understand $n_2$? Is it the number of regenerated buffers that pass the guard? Or is it the total number of prompts that yield good responses without running out of regenerations?

**Ethical Concerns:**

["NO or VERY MINOR ethics concerns only"]

**Final Justification:**

The paper tackles an important problem with solid execution - the motivation for decoding-time safety interventions is clear, the writing is good, and the experimental evaluation is thorough with proper baselines. The main concerns from my initial review were adequately addressed in the rebuttal: the authors agreed to cite relevant related work, and agreed to clarify the explanation of the "success rate" metric.

**Limitations:**

The authors should discuss the limitations of the method in the conclusion section. For example, the cost of evaluating with a guard model frequently, the cost of re-generation, and the extra latency for the end user. I also wonder whether this type of system if deployed could be adversarially exploited.

**Paper Formatting Concerns:**

- I'd suggest rephrasing the following sentence in the introduction, as it is hard to parse:
	- "As we studied in Section 2, the quality of responses that do not actually require intervention degrades significantly."
- Lines 68-69: typo. I think it could be fixed by adding "before and after, respectively".

**Quality:**

3

**Strengths And Weaknesses:**

Strengths
- Well motivated problem.
	- Avoiding harmful generations in LLM inference is an important problem. Most solutions are at the level of model training, rather than decoding methodologies. Achieving safe outputs via decoding strategies is well motivated.
- Well written paper.
	- I found the paper mostly clear, and easy to read.
- Thorough experimentation
	- The authors diligently evaluate a variety of decoding methods, including reasonable baselines including vanilla repeated sampling. The authors also evaluate multiple parameters of the method, and measure their effect in a controlled way.

Weaknesses
- Lack of discussion of limitations.
	- I think the conclusion could discuss the limitations of the method. For example, the cost of evaluating with a guard model frequently, the cost of re-generation, and the extra latency for the end user. I also wonder whether this type of system if deployed could be adversarially exploited.

- (Not really a weakness, but could cite and discuss Sharma et al 2025 (https://arxiv.org/abs/2501.18837)).

---

> ### Author Rebuttal · Authors · 2025-07-30
>
> **Weakness 1**: I think the conclusion could discuss the limitations of the method. For example, the cost of evaluating with a guard model frequently, the cost of re-generation, and the extra latency for the end user. I also wonder whether this type of system if deployed could be adversarially exploited.
>
> **Response**: We thank the reviewer for their insightful feedback and for highlighting these considerations. We will expand the discussion on limitations and potential vulnerabilities in the final manuscript.
>
> * Regarding the cost of re-generation and latency, **our work provides a detailed analysis of this trade-off**. We introduced the Average Wait Tokens (AWT) metric specifically to quantify this latency cost. Our experimental results demonstrate that this overhead is a meaningful exchange for substantial gains in intervention effectiveness and overall response quality. Crucially, it is important to note that our framework's intervention is targeted. This latency cost is incurred only on the small subset of queries flagged as unsafe. Since the vast majority of queries in a real-world setting are benign and require no intervention, the average, amortized overhead experienced by the end-user is minimal and **well within acceptable limits for practical deployment.**
>
> * Regading the Cost of Guard Model Evaluation: The cost of frequent evaluation is indeed a key factor. To address this, we will add a detailed discussion to our Limitations section on **practical strategies to mitigate this overhead**. These strategies include:
>     * Using smaller, specialized guard models: Employing a distilled or smaller classifier model instead of a large LLM can drastically reduce evaluation costs.
>     * Optimizing check frequency: Our current implementation already performs checks in batches (every b/2 tokens) rather than per-token to improve efficiency. This frequency can be further tuned.
>     * Utilizing prompt caching: Techniques like prompt caching can be used for the guard model to avoid re-computing for repeated prefixes.
> * Regarding the Potential for Adversarial Exploitation: We concur with the reviewer that adversarial robustness is a critical concern for any deployed safety system. Our framework's security is indeed linked to the robustness of its guard model. ***Our work's core contribution is the modular detect-rollback-intervene framework and the novel  Introspection method, which are designed to be decoupled from the specific guard model used***. This modularity is a strength, as our framework can readily integrate and benefit from more advanced, adversarially robust guard models. In the revised version, we will explicitly add this to our Limitations section. We will clarify that while the guard model is a potential attack vector, this challenge is orthogonal to our primary contribution, and we will call for more research into robust guard models that can be seamlessly integrated into frameworks like ours.
>  ----
> **Weakness 2**: (Not really a weakness, but could cite and discuss Sharma et al, 2025.
>
> **Response**: We sincerely thank the reviewer for pointing us to this highly relevant work by Sharma et al. (2025). We have read the paper and find it an important contribution to the field of ensuring the LLM safety, and we will certainly cite and discuss it in our Related Work section.
>
> ----
>
>
> **Question 1**: What is the guard model used in experiments?
>
> **Response**: Thank you for the question. The guard model used in our experiments is ***cais/HarmBench-Llama-2-13b-cls***. Due to space constraints in the main paper, we introduced this model in Appendix B. This model is the official safety classifier for the HarmBench [C1] benchmark, developed by Center for AI Safety (CAIS). It is also used in official academic competitions, such as the NeurIPS 2024 Trojan Detection Competition. This classifier has been rigorously evaluated and shown to **achieve over 95% correlation with human judgments on safety tasks**. Its performance in identifying harmful content has been documented to surpass other widely-used safety models, including GPT-4-as-a-judge and Meta's LlamaGuard.
>
> ----
> **Question 2**: Could you clarify success rate?  I found the explanation unclear. Does n1 increment any time the guard model flags a buffer? Does it increment for regenerated buffers? Or is it the number of prompts where there is at least one regeneration? This is unclear to me, and I don't understand how n1 could be "fixed for all methods", unless my last interpretation is correct. And similarly, how can I understand n2? Is it the number of regenerated buffers that pass the guard? Or is it the total number of prompts that yield good responses without running out of regenerations?
>
> **Response**: Thank you for raising this question, which highlights an important opportunity to improve the clarity of our definitions.  ***Your final interpretation is indeed correct***. As clarified in Appendix D.1,  $n_1$  represents the number of prompts for which at least one regeneration occurs (indicating a security risk during the generation process), and this value remains fixed across all methods. The quantity $n_2$ refers to the total number of prompts that require at least one regeneration but ultimately produce valid responses without exhausting the allowed regeneration attempts. We agree that the explanation in the main paper could be more explicit. In the final manuscript, we will revise Section 4.1 to align with the more detailed definitions provided in Appendix D.1, ensuring greater clarity for all readers.
>
> ----
> # References
>
> [C1] HarmBench: A Standardized Evaluation Framework for Automated Red Teaming and Robust Refusal. Mantas Mazeika, Long Phan, Xuwang Yin, Andy Zou, Zifan Wang, Norman Mu, Elham Sakhaee, Nathaniel Li, Steven Basart, Bo Li, David Forsyth, Dan Hendrycks

---

> > ### Comment · Reviewer_9AA7 · 2025-08-01
> >
> > Thank you for the replies.
> >
> > I think the work will be improved by the modifications suggested by the authors, in particular the clarification of success rate in the main body.

---

> > > ### Author Response · Authors · 2025-08-02
> > > **thank you for the positive feedback**
> > >
> > > Thank you for the positive feedback. We are glad that our clarifications, particularly regarding the success rate, were helpful. We look forward to incorporating all the suggested modifications into the final version of the paper.

---

### Author Response · Authors · 2025-08-09
**Final Summary of Author-Reviewer Discussion**

We sincerely thank all reviewers for an engaging and highly constructive discussion period. We are very grateful that our detailed rebuttals and new experiments seem to successfully address all initial concerns, resulting in positive final feedback from all reviewers. **We offer a brief summary of these outcomes below**:

* (Reviewer afd5; initial score: 3 - Borderline Reject): The reviewer's primary concerns were about technical novelty and generalization. After our rebuttal, which included new experiments on Llama 3.1 and the WildGuard benchmark, and a detailed discussion differentiating our work from the reasoning literature, the reviewer confirmed they were convinced. They stated they **"recognize its potential value for decoding-time safety alignment" and have updated their score.**

* (Reviewer UFMt; initial score: 2 - Reject): The reviewer's main concerns were about the novelty of the framework and the evaluation methodology. We provided detailed clarifications on the novelty of our system-level workflow and new experiments with multi-seed runs to prove the stability of our metrics. The reviewer confirmed that our framing of **novelty as "leveraging an existing mechanism for an entirely new purpose" was convincing and stated they will increase the score.**

* (Reviewer 9AA7 & vnyM; initial scores: 4 - Borderline Accept): These reviewers raised important questions regarding limitations, metric clarity, and evaluation details. We provided detailed answers and new statistical analyses. Both reviewers responded positively, with Reviewer 9AA7 noting the paper **"will be improved by the modifications"** and Reviewer vnyM confirming they **"remain positive about this paper."**

---

### Decision · Program_Chairs · 2025-09-17

**Decision:**

Accept (poster)

**Comment:**

This paper addresses the important problem of improving decoding-time safety alignment for LLMs by combining three elements: a guard model for real-time unsafe content detection, a rollback mechanism that corrects unsafe continuations by reverting to a safe buffer, and an introspection-based intervention where the model critiques and updates its own outputs. The approach is well-motivated, timely, and clearly presented, and the experimental results show that the framework can improve safety interventions with minimal latency while preserving output quality. Strengths include the novelty of integrating rollback with introspection, strong empirical results demonstrating effectiveness across benchmarks, and relevance to practical deployment where training-time alignment alone is insufficient. The main weaknesses are that the scope of evaluation is somewhat limited (e.g., more real-world adversarial safety scenarios would strengthen the case), and some methodological details (such as reliance on the chosen guard model and introspection prompt design) could be more fully analyzed. Overall, the work makes a meaningful contribution to decoding-time alignment methods, and I recommend acceptance as a poster (if the authors do all changes as promised).